# Rapid Synergistic Biofilm Production of *Pseudomonas* and *Candida* on the Pulmonary Cell Surface and in Mice, a Possible Cause of Chronic Mixed Organismal Lung Lesions

**DOI:** 10.3390/ijms23169202

**Published:** 2022-08-16

**Authors:** Pornpimol Phuengmaung, Jiradej Mekjaroen, Wilasinee Saisorn, Tanittha Chatsuwan, Poorichaya Somparn, Asada Leelahavanichkul

**Affiliations:** 1Department of Microbiology, Faculty of Medicine, Chulalongkorn University, Bangkok 10330, Thailand; 2Center of Excellence on Translational Research in Inflammation and Immunology (CETRII), Department of Microbiology, Faculty of Medicine, Chulalongkorn University, Bangkok 10330, Thailand; 3Center of Excellence in Systems Biology, Research Affairs, Faculty of Medicine, Chulalongkorn University, Bangkok 10330, Thailand

**Keywords:** *Pseudomonas aeruginosa*, *Candida albicans*, interkingdom biofilms, alginate, psl, acute pneumonia

## Abstract

Due to the possible co-presence of *Pseudomonas aeruginosa* and *Candida albicans* (the most common nosocomial pathogens) in lungs, rapid interkingdom biofilm production is possible. As such, PA+CA produced more dominant biofilms on the pulmonary epithelial surface (NCI-H292) (confocal fluorescent extracellular matrix staining) with dominant *psl* upregulation, as demonstrated by polymerase chain reaction (PCR), after 8 h of experiments than PA alone. With a proteomic analysis, rhamnosyltransferase RhlB protein (Psl-associated quorum-sensing protein) was found to be among the high-abundance proteins in PA+CA than in PA biofilms, supporting *psl*-mediated biofilms in PA+CA on the cell surface. Additionally, PA+CA increased supernatant cytokines (IL-8 and IL-13, but not TNF-α, IL-6, and IL-10) with a similar upregulation of *TLR-4*, *TLR-5*, and *TLR-9* (by PCR) compared with PA-stimulated cells. The intratracheal administration of PA+CA induced a greater severity of sepsis (serum creatinine, alanine transaminase, serum cytokines, and histology score) and prominent biofilms (fluorescent staining) with *psl* upregulation (PCR). In comparison with PA+CA biofilms on glass slides, PA+CA biofilms on biotic surfaces were more prominent (fluorescent staining). In conclusion, PA+CA induced Psl-predominant biofilms on the pulmonary cell surface and in mice with acute pneumonia, and these biofilms were more prominent than those induced by PA alone, highlighting the impact of *Candida* on rapid interkingdom biofilm production.

## 1. Introduction

Biofilms are microbial communities that are surrounded by heterogeneously complex extracellular polymeric substances (EPSs) resulting, in part, from the harsh micro-environment, caused by host immunity, limited nutrient availability, and competition from other microbes [1]. Quorum-sensing is one of the most potent molecules for the initiation of cell-to-cell communication within the microbial communities of microorganisms, followed by the synthesis of the functional molecules that have an impact on the different phenotypes of biofilms, such as the ability to attach to abiotic and biotic surfaces [2,3,4]. Because biofilm-residing microorganisms are protected from host immune responses and antibiotics, infections with biofilms are usually difficult to treat [5].

A mixed infection by different bacteria (polymicrobial infection) or organisms (interspecies infection) is well-known, as *Pseudomonas aeruginosa* and *Staphylococcus aureus* are the two of the most prevalent bacterial species in the lungs of cystic fibrosis [6]. Likewise, 20% of systemic *Candida albicans* infections are polymicrobial, with *Staphylococcus epidermidis* and *Staphylococcus aureus* being the first and third most common co-isolated organisms, respectively, with a strong interspecies biofilm formation [7,8]. Moreover, an enhanced severity of infection with the co-presence of *C. albicans* and *Klebsiella pneumoniae* has been reported [9]. However, *Pseudomonas aeruginosa* and *Candida albicans* are two of the most common opportunistic pathogens that usually colonize numerous sites of the human body, resulting in several diseases, including burn wounds, contaminated catheters, and infections in several organs (the genitourinary tract, gut, and lung). Interestingly, the co-presence of *Pseudomonas* and *Candida* induces prominent interkingdom biofilms [10,11,12,13] with more severe infections [14,15,16] as indicated in lungs with cystic fibrosis and ventilation-associated pneumonia (VAP) [17,18,19,20,21], which is partly due to the potent *Pseudomonas* quorum-sensing proteins (growth promotion) and elastases (disruption in the tight junctions of host cells) [22,23]. Additionally, some molecules from *Candida* spp. also facilitate the biofilm productivity of *Pseudomonas*, as the addition of *Candida* spp. into preformed *Pseudomonas* biofilms increases biofilm production more than does the initial simultaneous incubation of *Pseudomonas* with *Candida* [13].

Because the presence of *C. albicans* in the human pulmonary system is frequently reported [24] and because of the high prevalence of nosocomial pneumonia from *Pseudomonas* spp. [25], the co-existence of *Pseudomonas* and *Candida* in pulmonary lesions and their synergy in biofilm production is possible. However, most of the previous publications on the synergy of *Pseudomonas–Candida* biofilms are conducted using abiotic surfaces (glass slides, plastics, and catheters) [13]. While *Pseudomonas* biofilms on abiotic surfaces consist of an alginate-predominant matrix, biofilms on the bronchial airway are composed of proteins mainly from the polysaccharide synthesis locus (Psl) gene [26,27], implying that biofilm properties depend on surface characteristics. Thus, studies of biofilms on biotic surfaces might better resemble patients’ conditions. 

Moreover, in vivo biofilm formation is mostly demonstrated in chronic infections, especially *Pseudomonas* and *Candida*, despite the rapid formation of biofilms (within 1–2 days) after in vitro incubation and the common co-presence of both organisms [13]. Hence, the co-presentation of both organisms on the biotic surface of the pulmonary system for a short period might cause chronic interkingdom lesions, and the understanding of this pathophysiology is beneficial for the treatment of *Pseudomonas–Candida* infections in several diseases (cystic fibrosis, VAP, and other nosocomial pneumonias). Therefore, we tested the influence of the co-presence of *Pseudomonas–Candida* on a human pulmonary mucoepidermoid carcinoma cell line (NCI-H292) and on glass slides, together with a mouse model of aspiration pneumonia. 

## 2. Results

### 2.1. Prominent Mucoid Biofilms from Pseudomonas plus Candida on Human Pulmonary Cells

Due to the lack of data on interkingdom *Pseudomonas–Candida* biofilms on pulmonary cell surfaces, biofilms from mixed organisms and *Pseudomonas* alone were stained for extracellular matrix, bacterial DNA, and fungi (Figure 1A,B). At 24 h after inoculation, there were more prominent *Pseudomonas–Candida* biofilms (PA+CA) with a higher abundance of bacteria and fungi than *Pseudomonas* alone (PA) biofilms (Figure 1A,B). Additionally, PA+CA and PA alone reduced the number of pulmonary cells as indicated by an MTT assay (this was more prominent in the PA+CA group than in the PA alone group) (Appendix A) due to *Pseudomonas* cytotoxicity [6]. There was an increase in the intensity of yellow fluorescence (Figure 1A, middle), which might represent elevated cell tight junctions and bacterial DNA abundance; however, bacterial abundance, determined by culturing, was higher in the PA+CA biofilms than in the PA alone biofilms (Appendix A). Although the enhanced blue color in the PA+CA biofilms than that in the PA alone biofilms (Figure 1A right) might be due to an increase in both pulmonary and yeast cells, the fungal abundance in the PA+CA biofilms was higher than that in the PA alone biofilms (Appendix A). The cell injury, as determined by the loss of cell tight junctions (ZO-1), was higher in the PA alone biofilms than in the PA+CA biofilms, as the linear yellow staining color in the PA+CA biofilms was more prominent than that in the PA alone biofilms (Figure 1B), supported by the higher number of dead cells observed in the PA alone biofilms than in the PA+CA biofilms using the MTT assay (CA alone did not induce cell death) (Appendix A). Notably, the incubation of *Candida* alone did not induce the formation of biofilms on the pulmonary cells (Appendix A). It is possible that the better conditions of the biotic surface in the PA+CA group than those of the unhealthy surface in the PA alone group were partly responsible for the more prominent biofilms in the PA+CA condition.

At 8 h post-incubation, PA+CA induced a lower expression of *pslA* (a gene required for Psl production) with higher *mucA* and *mucB* (negative regulators of alginate) but similar *algD* (a gene required for alginate production) than PA alone (Figure 2A–D). After 24 h of incubation, the expression of *mucA*, but not other genes, was higher in the PA+CA group than in the PA alone group (Figure 2A–D). These data suggest a possible difference between the biofilm properties of PA+CA and PA alone through the more prominent *psl*-mediated biofilm components, with a more dominant alginate negative regulator (*mucA*) in PA+CA biofilms than in PA alone biofilms on the lung epithelial cell surface [28,29]. 

Additionally, the differences between both groups were demonstrated by a proteomic analysis based on the fold change of the proteins in the PA+CA biofilms compared with that of the proteins in the PA alone biofilms, as indicated by a volcano plot (Figure 2E), and only 76 proteins were demonstrated to be higher in the PA+CA biofilms than in the PA alone biofilms (Figure 2F) using the protein biological process based on the PANTHER classification (http://www.pantherdb.org; accessed on 15 June 2022). These proteins included the proteins for rhamnosyltransferase RhlB (Psl-associated quorum-sensing) and iron-translocating oxidoreductase complex subunit C (RnfC) (Figure 2G). Meanwhile, there were 281 proteins that were expressed in the PA+CA group at a lower level than in the PA alone group, including the proteins for alginate-mediated biofilms, transcriptional regulatory protein (AlgP), alginate biosynthesis transcriptional regulatory protein (AlgB), phosphomannomutase/phosphoglucomutase (AlgC), and *Pseudomonas* pathogenesis (Tol–Pal system protein; TolB) (Figure 2H). Notably, the initial abundance of bacteria in the PA+CA group was half of that in the PA alone group (organismal abundance between PA+CA and PA alone groups was controlled), and most of the bacterial proteins in the PA biofilms were higher than those in the PA+CA biofilms. The higher proteins in the PA+CA group than those in the PA alone group are the interesting *Candida*-inducing *Pseudomonas* proteins.

### 2.2. The Pulmonary Responses against Interkingdom Pseudomonas–Candida Biofilms

Because (i) the better pulmonary cell tight junction in the PA+CA group than that in the PA alone group (Figure 1B, the linear yellow color) might be due to the higher *psl*-mediated *Pseudomonas* exopolysaccharides (Figure 2A–D) (possibly induced by fungal ethanol [30]) and because (ii) pulmonary impacts from *psl*-mediated lung responses [26,31] with *Pseudomonas–Candida* competition is possible [32], the cell responses against PA+CA and PA alone might be different. Indeed, PA+CA biofilms induced higher proinflammatory responses in pulmonary epithelial cells, mostly at 8 h post-stimulation as indicated by supernatant IL-8 and IL-13, but not TNF-α, IL-6, or IL-10 than PA alone biofilms (Figure 3A–E). Notably, IL-8 and IL-13 are cytokines for chemotaxis and pulmonary mucosal inducers (inflammation, mucus hyperproduction, goblet cell hyperplasia, and subepithelial airway fibrosis), respectively [33,34]. Meanwhile, PA biofilms increased supernatant levels of IL-8 and IL-10, but did not increase the levels of other cytokines, when compared with CA biofilms (CA induced the lowest cytokine levels among these triggers) (Figure 3A–E). In parallel, similar *Toll-like receptor 4* (TLR-4) upregulation was found in pulmonary epithelial cells after PA alone and PA+CA (higher than CA activation) (Figure 3F), implying a more potent TLR-4 inducer of LPS from *P. aeruginosa* than the stimulation from *Candida* antigens [35]. However, the upregulation of *TLR-5* was the most prominent in PA+CA activation (at 8 h after incubation) than in PA or CA alone (Figure 3G), supporting the more potent TLR-5 activation by *Pseudomonas* flagella than *Candida* antigens [36]. Furthermore, *TLR-9* upregulation was similar between PA+CA and PA alone groups, and this was higher than in the CA group (Figure 3H), possibly due to the higher cytosine–guanine dinucleotide (CpG) DNA in bacteria than in fungi [37]. Hence, the co-presence of *Pseudomonas* with *Candida* induced the most potent hyper-inflammation, as indicated by cytokines (IL-8 and IL-13), partly though the activation of TLR-4, TLR-5, and TLR-9. 

### 2.3. Biofilm Formation in the Lungs of Acute Pneumonia, a Possible Initiator of Chronic Lesions

Because of the initial epithelial adhesion property of psl-mediated biofilms in mucoid *Pseudomonas* being more potent than that of alginate-predominant biofilms [26] and because of *Candida*-induced *psl* upregulation (Figure 2A), we hypothesized that *Pseudomonas* with *Candida* could rapidly produce biofilms and tested it in vivo (Figure 4A). At 24 h post-infection, sepsis severity was more prominent in the PA+CA group than in the PA alone group, as indicated by acute kidney injury (serum creatinine), liver damage (alanine transaminase), serum cytokines (TNF-α and IL-6, but not IL-10 or IL-1β), bacteremia, lung injury score, and the expression of biofilm-associated genes (*pslA*, *mucA*, and *algD*) (Figure 4B–H). There was no fungemia in the PA+CA group (data not shown), and the more prominent bacteremia in the PA+CA group than that in the PA alone group might be due to the higher bacterial abundance in the PA+CA group than that in the PA alone group on the pulmonary surface (Appendix A). The *psl* upregulation in the PA+CA group supports the enhanced *Pseudomonas* biofilms by *Candida* on the pulmonary cell line at 24 post-incubation *in vitro* (Figure 1A,B). Despite the limited data on biofilms in acute pneumonia, the fluorescent staining of mouse lung tissue demonstrated the dominant extracellular matrix along with higher bacterial and fungal DNA in the PA+CA group when compared with those in the PA alone group (Figure 5A,B). There was a higher abundance of bacterial DNA in the PA+CA group at 24 h of the pneumonia model (Figure 5A,B), although the initial administration of fewer bacterial abundances in the PA+CA group than in the PA alone group was used as a control for the equivalent microbial abundance between groups (see Materials and Methods Section). Notably, the possible protective pulmonary tight junction of the PA+CA group on the cell line (Figure 1B, yellow) was not obvious in the mice (Figure 5B, yellow), suggesting a difference between the in vitro and in vivo conditions. Additionally, the biofilm abundance of the PA+CA group on the mouse lung tissue as indicated by extracellular matrix staining (red) (Figure 5A,B) was similar to the in vitro biofilms on the cell line (Figure 1A,B), which were clearly more prominent than the biofilms on the abiotic surface (glass slide) (Figure 6A,B). Indeed, the fluorescent intensities of the extracellular matrix and the microbial DNA (bacteria and fungi) of the PA+CA biofilms on the biotic surfaces were higher than those on the abiotic glass slides (Figure 6A,B). Meanwhile, the abundance of the PA alone biofilms on abiotic glass slides was not different to that on biotic surfaces (Figure 6A,B). These data indicate the possible impact of biotic surface on biofilm formation and rapid biofilm formation, especially on bacterial–fungal interkingdom biofilms. The abundance of *Candida* in the respiratory system of patients with pneumonia might be associated with the risk of biofilm formation and chronic infection. More studies on these topics are interesting

## 3. Discussion

Although the physical and molecular interactions between *P. aeruginosa* and *C. albicans* have been extensively studied [38,39], the data on *Pseudomonas–Candida* interkingdom biofilms are still limited. The co-existence of *P. aeruginosa* and *C. albicans* on catheters enhances biofilm thickness via the alginate-related extracellular matrix [13], whereas the presence of *Pseudomonas–Candida* colonization on airway epithelial cells might be different. Indeed, *Pseudomonas* produces some molecules (such as phenazines) that induce ethanol from *Candida*, and ethanol facilitates the *Pseudomonas* extracellular matrix through the psl operon as a positive feedback loop [13], perhaps in order to adapt to the micro-environment. Because (i) the prominent biofilm production of PA+CA over PA alone might be due to the different biofilm properties and because of (ii) the importance of several genes for *Pseudomonas* biofilm production, especially for biofilms from the polysaccharide synthesis locus (Psl) versus the alginate-mediated pathway [22,25], the expression of some genes was explored. *Pseudomonas* biofilm-producing genes (alginate, psl, and pel) are important *Pseudomonas* virulent factors [40]. As such, alginate is a negatively charged acetylated polymer containing nonrepetitive β-1,4-linked l-guluronic and d-mannuronic acids used for inhibiting phagocytosis and enhancing reactive oxygen species in host cells [41,42,43,44]. Additionally, Psl is a neutral-branched pentasaccharide with a ratio of 1:1:3 of d-glucose, d-rhamnose, and d-mannose, and it mediates attachment to lung epithelial cells, promoting a pro-inflammatory response and host cell damage [31,45]. Meanwhile, Pel is a positively charged polysaccharide containing acetylated 1→4 glycosidic branches of N-acetylgalactosamine and N-acetylglucosamine, which stabilize the biofilm structure [46,47]. In parallel, *Candida* biofilms, polysaccharide-rich extracellular matrices [48], especially in the hyphae form, can be facilitated by yeast-to-hyphae accelerators of *Pseudomonas* (quorum-sensing molecules) together with hyphae-to-yeast inhibitors (such as farnesol) from the *Candida* themselves [49,50]. Different from *Pseudomonas–Candida* alginate-dominant biofilms on catheters [13], the biotic surfaces upregulated *pslA* that was more profound than the *Pseudomonas* alone biofilms, implying an important role of the lung cell surface on interkingdom biofilms. In the proteomic analysis, *Pseudomonas–Candida* proteins demonstrated the prominent proteins for iron-translocating oxidoreductase complex subunit C (RnfC) and rhamnosyltransferase RhlB (Psl-related quorum-sensing protein) with lower alginate-associated proteins (AlgP, AlgB, and AlgC) when compared to *Pseudomonas* alone, suggesting that the presence of *Candida* possibly led to the competition for iron resources (a source of nutrition) and that it induced Psl-dominated biofilms, which effectively attach to epithelial cells with lower cell responses than alginate-based biofilms [51]. 

Because the epithelial barrier integrity [32,52] is important for the first defense against pathogens [53,54] and because the loss of ZO-1 tight junction molecules in human lungs with *P. aeruginosa* infection has also been previously demonstrated [55,56], studies of biofilms on epithelial cell surfaces might better resemble the clinical situation. Here, there were increased biofilms in *Pseudomonas–Candida* compared to *Pseudomonas* alone on both the cell line and lung tissue, with an increased cell abundance of both bacteria and fungi, which was more prominent than the abundance on the glass slides, implying that the biotic surface has an impact on microbial growth and biofilm production. Despite *Pseudomonas–Candida* seeming to protect the pulmonary tight junction better than *Pseudomonas* alone (possibly for a better biofilm structure), the increased bacterial abundance of the *Pseudomonas–Candida* group might have resulted in higher bacteremia and more severe sepsis in our mouse model. The prominent *psl* upregulation, with the reduced expression of alginate-associated genes, was indicated by the high *mucA* (an alginate regulator) and low *algD* (an alginate promotor) of *Pseudomonas–Candida* compared to *Pseudomonas* alone, and this was similarly demonstrated both on pulmonary cells and in mouse lungs, despite the possible better nourishment and host–organism ratio in mice. Regarding the host responses against biofilms, *Pseudomonas–Candida* biofilms induced more severe responses than *Pseudomonas* alone biofilms through several well-known signals, including TLR-4, TLR-5, and TLR-9 from the activation by LPS, flagella, and microbial DNA [57,58,59]. Interestingly, microbial dissemination was demonstrated only from bacteremia but not fungemia, perhaps the yeast cells were too large to pass through the damaged pulmonary barrier, and the more severe sepsis in the *Pseudomonas–Candida* group might be mainly due to the higher bacterial abundance in the blood than in the mice with *Pseudomonas* alone. Despite the similar bacterial abundance demonstrated between *Pseudomonas–Candida* biofilms and *Pseudomonas* alone biofilms on the abiotic surface (glass slide) in our previous report [13], the microbial abundance (both bacteria and fungi) of *Pseudomonas–Candida* biofilms was higher than that of *Pseudomonas* alone biofilms, highlighting that host cells have an impact on microbial growth (in a similar way to microbial contamination on the cell culture) [60].

Interestingly, the treatment of *bacterial–Candida* interspecies biofilms is challenging, as the effect of antibacterials alone seems to be decreased in dual-species biofilms compared with bacteria alone biofilms [61]. Meanwhile, in *Pseudomonas–Candida* biofilms, a single treatment with an anti-fungal drug (fluconazole) alone synergistically with *Pseudomonas* on the biofilms more effectively reduces *Candida* burdens, partly through iron sequestration by *P. aeruginosa* [11]. These data indicate that anti-fungals might be beneficial in the treatment of *bacterial–Candida* interspecies biofilms. Moreover, quorum-quenching (quorum-sensing blockage) and anti-virulence compounds (inhibitors against the virulent factors that develop from dual biofilms) are new, interesting strategic treatments of inter-species biofilms [30]. More studies on this topic are required.

In our acute pneumonia model, the biofilms were surprisingly rapidly produced, which, in parts, was associated with the more severe antibiotic resistance and microbial persistence in the respiratory tract with the co-infection of *Pseudomonas–Candida* [62,63]. In this regard, the studies on biofilms, which currently are mostly only being conducted in some time-consuming chronic infection models, could be conducted in acute infection models (especially for the early phase of biofilms) and may rapidly expand the knowledge on the topic. However, chronic infection models are still necessary for the evaluation of the progression from the acute phase to the chronic phase of biofilms. For clinical translation, we propose that *Candida* abundance in the pulmonary tract of patients with *Pseudomonas* pneumonia might be associated with rapid biofilm production, treatment difficulties, and progression into a chronic form of infection. A further exploration of *Candida* abundance in lungs and pneumonia is warranted.

## 4. Materials and Methods

### 4.1. Microbial Preparation and Biofilms on Human Cells

*P. aeruginosa* and *C. albicans* were isolated from the endotracheal specimens of patients from the King Chulalongkorn Memorial Hospital, Bangkok, Thailand, following the protocol (MDCU-IBC001/2022) approved by the ethical institutional review board, Faculty of Medicine, Chulalongkorn University, according to the Declaration of Helsinki, with written informed consents. Human mucoepidermoid carcinoma (NCI-H292) cells were grown and maintained in Dulbecco’s modified Eagle medium (DMEM; Life Technologies, Carlsbad, CA, USA), supplemented with 10% heat-inactivated fetal bovine serum (FBS; Life Technologies) and 1% penicillin/streptomycin in a humidified 5% CO_2_ incubator at 37 °C. Both *P. aeruginosa* and *C. albicans* were grown in tryptic soy broth (TSB; Difco™, Becton, NJ, USA) to induce the early stationary phase, which is appropriate for biofilm formation, and they were washed and resuspended in fresh prewarmed completed DMEM containing 10% FBS to adjust the turbidity of 1 × 10^6^ CFU/mL. The suspensions (*P. aeruginosa* plus *C. albicans* or each organism alone) were added to 12-well polystyrene plates with 1 × 10^6^ confluent NCI-H292 cells at the multiplicity of infection (MOI) between organisms per cells at 1:100 on a biotic surface or on the polystyrene well with a round glass coverslip in the bottom (abiotic surface) in order to evaluate biofilm formation. To control the number of organisms in each condition, 1 × 10^6^ CFU/mL of the preparation of *P. aeruginosa* alone (PA) or *C. albicans* alone (CA) was used, while 0.5 × 10^6^ CFU/mL of *Pseudomonas* and *Candida* preparations was combined to use for the preparations of *Pseudomonas* plus *Candida* (PA+CA).

Biofilms were grown at 37 °C with 5% CO_2,_ and the culture medium was changed every 12 h. The viability of NCI-H292 in these biofilms was measured using an MTT (3-[4,5-dimethylthiazol-2-yl]-2,5 diphenyltetrazolium bromide) assay (Invitrogen, Waltham, MA, USA) according to the manufacturer’s procedures. Briefly, the 24 h activated cells were incubated with 0.5 mg/mL of MTT solution for 2 h at 37 °C in the dark before the removal of MTT, diluted with dimethyl sulfoxide (DMSO) (Thermo Fisher Scientific, Waltham, MA, USA), and measured using a Varioskan Flash microplate reader at an absorbance of optical density of 570 nm. Supernatants were quantified for cytokine measurement using an enzyme-linked immunosorbent assay (ELISA; Invitrogen). The ability of biofilm production on in vitro (biotic and abiotic surfaces) and in vivo the pulmonary lesions prepared in Tissue-Tek O.C.T Compound (Sakura Finetek, Torrace, CA, USA) (details later) was evaluated by employing confocal microscopy using ZEISS LSM 800 (Carl Zeiss, Jena, Germany) with the following antibodies: (i) extracellular matrix using Alexa FluorTM647-tagged Concanavalin A (Invitrogen) (red color of AF647), (ii) bacterial DNA by SYTO9 color (Invitrogen) (yellow color of AF488), (iii) pulmonary cell borders through tight junction zonula occluden-1 (ZO-1) (Thermo Fisher Scientific) with the goat anti-rabbit secondary IgG (Cell Signaling Technology, Danvers, MA, USA) (yellow color of AF488), (iv) Calcofluor white of fungal stain (Sigma-Aldrich, St. Louis, MI, USA) (blue color of AF460), and (v) nuclei of pulmonary cells by DAPI (4′,6-diamidino-2-phenylindole) color (Cell Signaling Technology) (blue color of AF460). Notably, AF488 yellow was used for both pulmonary tight junctions and bacterial DNA, while AF460 blue was stained for the nuclei of both human cells and fungi due to technical limitations. However, the different characteristics (shapes and sizes) between tight junctions and bacterial DNA, and the dissimilar nucleus sizes between human cells and fungi were used for differentiation.

### 4.2. Quantitative Real-Time Polymerase Chain Reaction

Several molecules associated with *P. aeruginosa* biofilms with and without *C. albicans* on NCI-H292 cells (inflammation and biofilm formation) were evaluated using quantitative real-time polymerase chain reaction (qRT-PCR). Total RNA was extracted by TRIzol reagent (Thermo Fisher Scientific), converted by reverse transcriptase (RevertAid First Stand cDNA), and conducted qRT-PCR using Applied Biosystems QuantStudio 6 Flex Real-Time PCR System (Applied Biosystems, Warrington, UK) with SYBR^®^ Green PCR master mix. The housekeeping gene *16S rRNA* was used to normalize the transcriptional levels of target genes from *P. aeruginosa*, and *GADPH* was used to normalize the gene expression in NCI-H292 cells, with the comparative cycle threshold against the expression. The list of primers for PCR is presented in Table 1.

### 4.3. Proteomic Analysis

Due to the biofilm production of *C. albicans* being less than that of *P. aeruginosa* [9], the pulmonary cells (NCI-H292) with biofilms from *P. aeruginosa* plus *C. albicans* or *P. aeruginosa* alone were used for a proteomic analysis. Notably, the initial abundance of *P. aeruginosa* in the *Pseudomonas* alone (PA) group was approximately 2-fold higher than that in the *Pseudomonas* plus *Candida* (PA+CA) group due to the protocol to ensure a similar total organismal abundance between both conditions (PA vs. PA+CA). Then, acetone precipitation of proteins was performed, and the pellet was resuspended with 8 M urea in 50 mM Tris-HCl (pH 8) to estimate protein concentration using a BCA protein assay (Thermo Fisher Scientific). Subsequently, the dissolved proteins were incubated with dithiothreitol (DTT), alkylated by iodoacetamide (IAA), digested with trypsin, inhibited the digestion by trifluoroacetic acid (TFA), and lyophilized by SpeedVac centrifugation (Thermo Fisher Scientific). After that, the peptides were analyzed using an EASY-nLC1000 system coupled to a Q-Exactive Orbitrap Plus mass spectrometer (Thermo Fisher Scientific) equipped with a nano-electrospray ion source (Thermo Fisher Scientific) at a flow rate of 300 nL/min, followed by a linear gradient from 5% to 40% acetonitrile in 0.1% formic acid for 60 min, with 40–95% acetonitrile in 0.1% formic acid for 30 min. Fractioned peptide samples were carried out with a liquid chromatography-mass spectrometer (LC-MS) using a 10 data-dependent acquisition method with an MS scan range of 350–1400 *m*/*z* accumulated at a resolution of 70,000 full widths at half maximum (FWHM) followed by a resolution of 17,500 FWHM. The normalized collision energy of higher energy collisional dissociation (HCD) was controlled at 32%. Precursor ions with unassigned charge states of +1 or those greater than +8 were excluded, and the dynamic exclusion time was set to 30 sec. Data acquisition and interpretation were monitored using Proteome DiscovererTM software 2.1 (Thermo Fisher Scientific). Peptide spectra were evaluated using the SEQUEST-HT search engine against *P. aeruginosa* (strain ATCC 15692/DSM 22644/CIP 104116/JCM 14847/LMG 12228/1C/PRS 101/PAO1) Swiss-Prot Database (1435 proteins, March 2022). The following parameters were set for the search: (i) digestion enzyme: trypsin; (ii) maximum allowance for missed cleavages: 2; (iii) maximum of modifications: 4; (iv) fixed modifications: carbamidomethylation of cysteine (+57.02146 Da), as well as light and medium dimethylation of N termini and lysine (+28.031300 and +32.056407 Da); and (v) variable modifications: oxidation of methionine (+15.99491 Da). A biological process analysis of the host proteins was performed using the PANTHER (http://www.pantherdb.org/; accessed on 15 June 2022) program. Mass spectrometry proteomics data were determined and submitted to the ProteomeXchange Consortium via PRIDE (http://www.proteomexchange.org, accessed on 15 June 2022) with the data set identifier PXD034528.

### 4.4. Animals and Animal Model

Male, 12-week-old C57BL/6 mice, purchased from Nomura Siam International (Pathumwan, Bangkok, Thailand), were intratracheally administered with 0.2 mL of *P. aeruginosa* (1 × 10^8^ CFU/mL) or P. aeruginosa (0.5 × 10^8^ CFU/mL) with C. albicans (0.5 × 10^8^ CFU/mL) before sacrificing by cardiac puncture under isoflurane anesthesia with sample collections at 24 h post-administration. For a histological analysis, paraffined-embedded lungs were fixed with 10% neutral-buffered formalin before the processes with Hematoxylin and Eosin (H&E) staining, and the lung injury was evaluated at 200× magnification in 10 randomly selected fields for each animal. The lung injury score was determined based on desquamation, dystelectasis/atelectasis, congestion, interstitial thickness, infiltration, and bronchial exudate with a modified lung injury score [24] as follows: 0, no injury; 1, minimal/discrete; 2, mild; 3, moderate; and 4, severe. For bacteremia and fungemia, 10 μL of mouse blood was spread onto tryptic soy agar (TSA; Difco™, Becton, NJ, USA) and Sabouraud dextrose agar (SDA; Oxoid, Cambridge, UK) and incubated at 37 °C for 24 h before colony enumeration.

### 4.5. Statistical Analysis

Data are presented as mean ± standard error (SE). The differences between groups were examined using a one-way analysis of variance (ANOVA) with Tukey’s analysis or Student’s *t*-test for comparisons of multiple groups or two groups, respectively. Log-rank test was used for a survival analysis. The time-point data were analyzed using a repeat measured ANOVA. SPSS 11.5 software (SPSS, Chicago, IL, USA) and GraphPad Prism version 8.0 were used for all statistical analyses. A *p*-value of <0.05 was considered statistically significant.

## 5. Conclusions

In conclusion, in this study, the *Pseudomonas–Candida* biofilms on biotic surfaces were more *psl* predominant than the mixed-organism biofilms on abiotic surfaces. Moreover, the *Pseudomonas–Candida* biofilms were more profound than the monomicrobial *Pseudomonas* biofilms, highlighting that the impact of *Candida* and the attachment to host cells enhance *Pseudomonas* biofilm production.

## Figures and Tables

**Figure 1 ijms-23-09202-f001:**
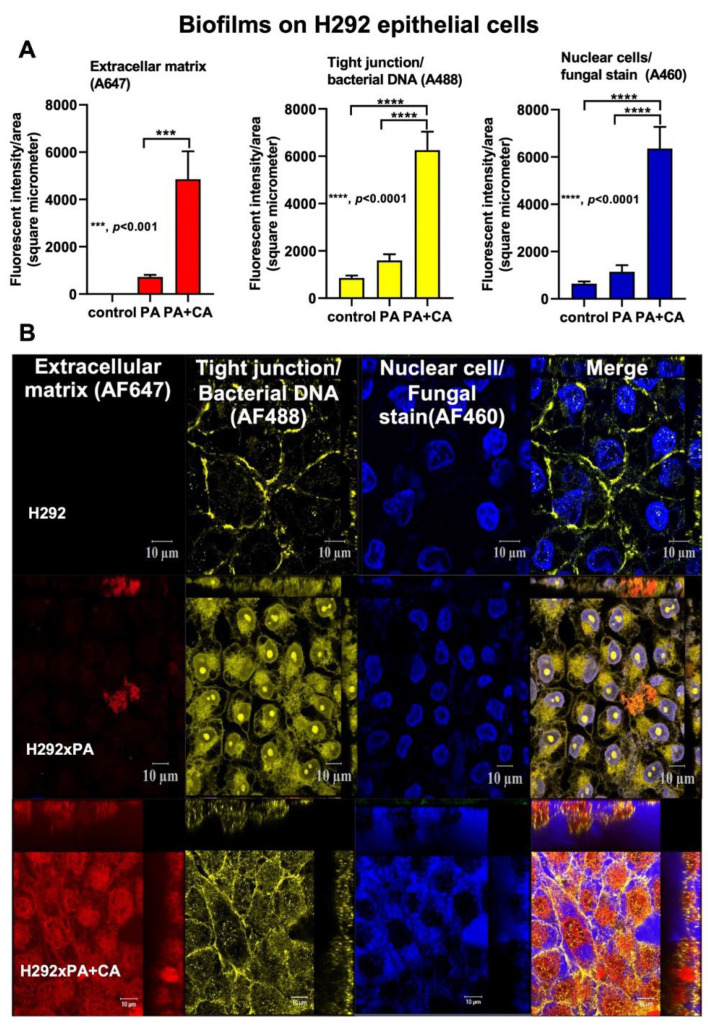
The characteristics of biofilms on pulmonary epithelial cells (NCI-H292) from *Pseudomonas* alone (PA), *Pseudomonas* with *Candida* (PA+CA), and control after staining for extracellular matrix (AF467, red), cell tight junction, bacterial DNA (AF488, yellow), and nuclei of host cells and fungi (AF460, blue) as indicated by fluorescent intensities (**A**), and the representative pictures (**B**) (independent experiments were performed in triplicate). Despite being the same color, the linear feature of the tight junction and the smaller size of the fungal nuclei compared to the host are used for differentiation (see text).

**Figure 2 ijms-23-09202-f002:**
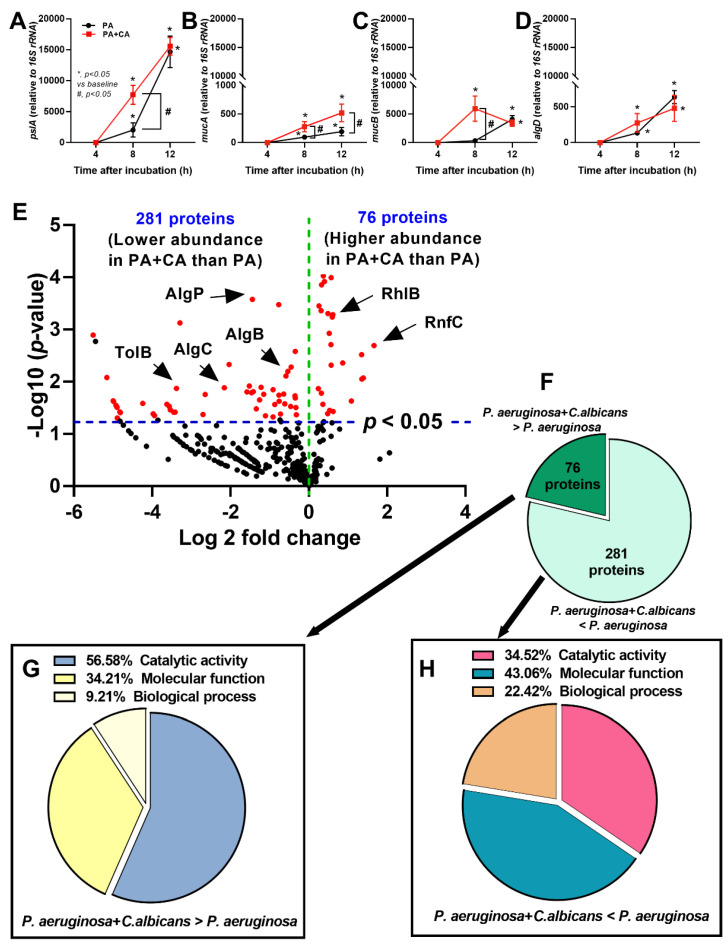
The characteristics of gene-associated biofilms using polymerase chain reaction (PCR) on pulmonary epithelial cells (NCI-H292) from *Pseudomonas* alone (PA) and *Pseudomonas* with *Candida* (PA+CA) are demonstrated (**A**–**D**) (independent experiments were performed in triplicate). A proteomic analysis of protein from PA+CA in relative to the protein from PA group is indicated by a volcano plot (**E**), and proteins with a higher abundance in PA+CA (*P. aeruginosa* + *C. albicans* > *P. aeruginosa*) or a lower abundance (*P. aeruginosa* + *C. albicans* < *P. aeruginosa*) with the functional characteristics of proteins from each group (**F**–**H**) are demonstrated (five samples per group for the proteomic analysis). AlgB, AlgC, and AlgP, proteins used for alginate-based biofilms; TolB, a part of the Tol–Pal system protein on bacterial cell membrane; rhamnosyltransferase RhlB, Psl-associated quorum-sensing protein); and RnfC, iron-translocating oxidoreductase complex subunit C.

**Figure 3 ijms-23-09202-f003:**
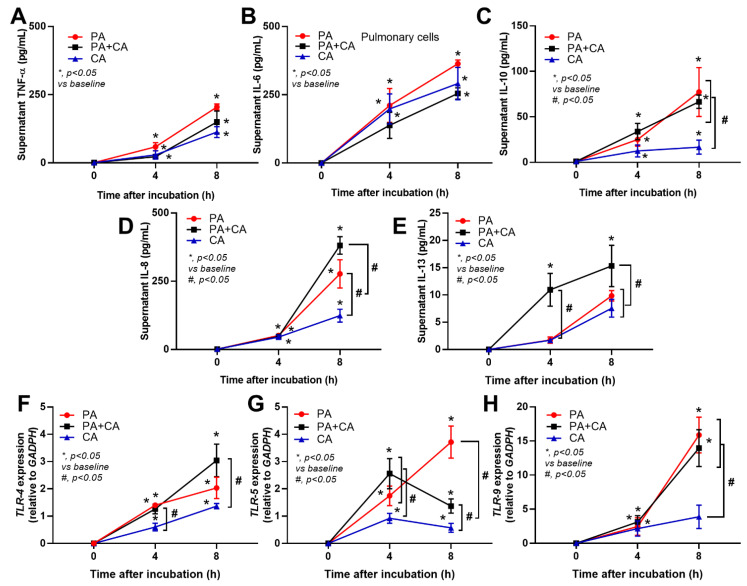
The responses of the pulmonary epithelial cells (NCI-H292) after activation by *Pseudomonas* alone (PA), *Candida* alone (CA), and *Pseudomonas* with *Candida* (PA+CA), as indicated by supernatant cytokines (TNF-α, IL-6, IL-10, IL-8, and IL-13) (**A**–**E**), along with the gene expression of *TLR-4*, *TLR-5*, and *TLR-9* using polymerase chain reaction (PCR) (**F**–**H**) (independent experiments were performed in triplicate).

**Figure 4 ijms-23-09202-f004:**
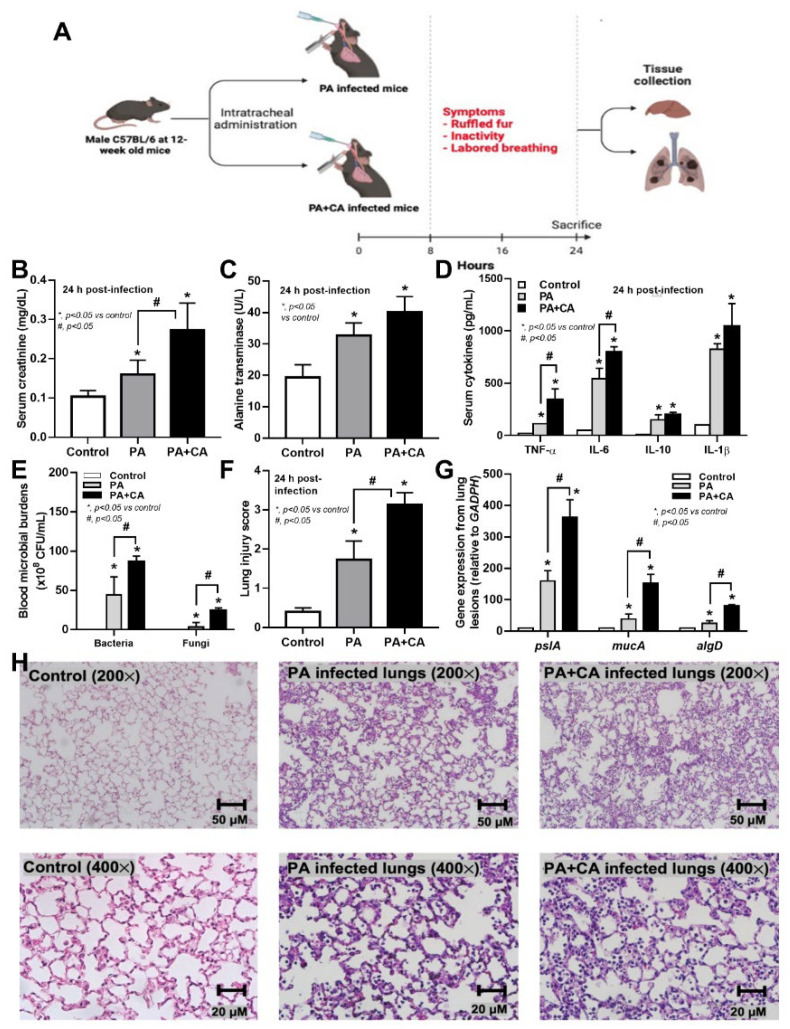
The experimental schema (**A**) was created using BioRender (http://app.biorender.com/) on 25 June 2022. The characteristics of sepsis in mice with pneumonia after administration of *Pseudomonas* alone (PA), *Pseudomonas* with *Candida* (PA+CA), or control, as indicated by serum creatinine (**B**), liver enzyme (**C**), serum cytokines (TNF-α, IL-6, IL-10, and IL-1β) (**D**), bacteremia burdens (**E**), lung injury score (**F**), and the expression of biofilm-associated genes (pslA, mucA, and algD) (**G**), and the representative pictures of lung lesions (**H**) (*n* = 6–7 per group).

**Figure 5 ijms-23-09202-f005:**
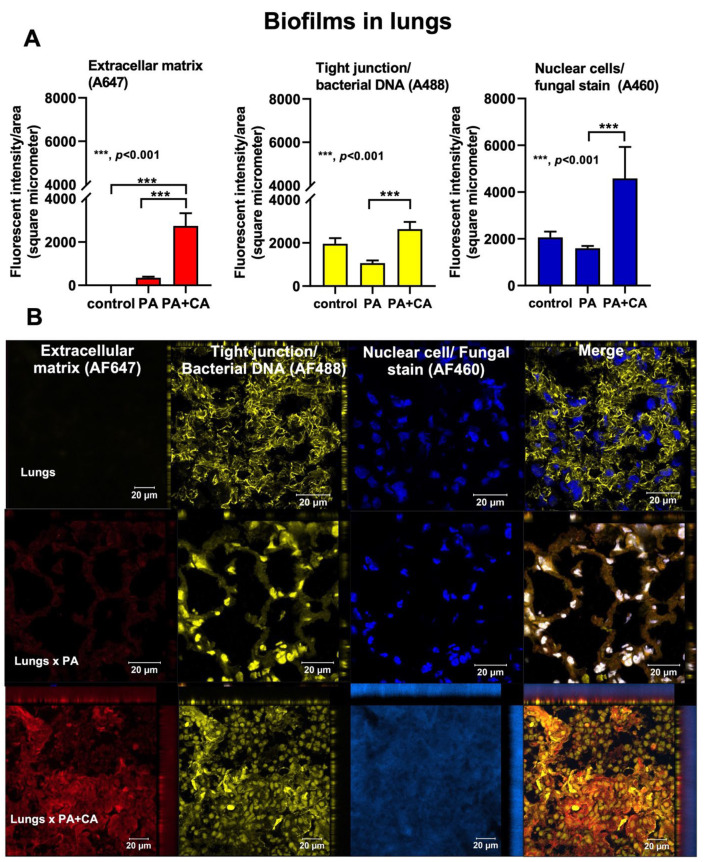
The characteristics of biofilms on mouse lung tissue after the administration of *Pseudomonas* alone (PA), *Pseudomonas* with *Candida* (PA+CA), or control after staining for extracellular matrix (AF467, red), cell tight junction, bacterial DNA (AF488, yellow), and nuclei of host cells and fungi (AF460, blue) as indicated by fluorescent intensities (**A**) and the representative pictures (**B**) (*n* = 6–7 per group). Despite being the same color, the linear feature of the tight junction and the smaller size of fungal nuclei compared to the host are used for differentiation (see text).

**Figure 6 ijms-23-09202-f006:**
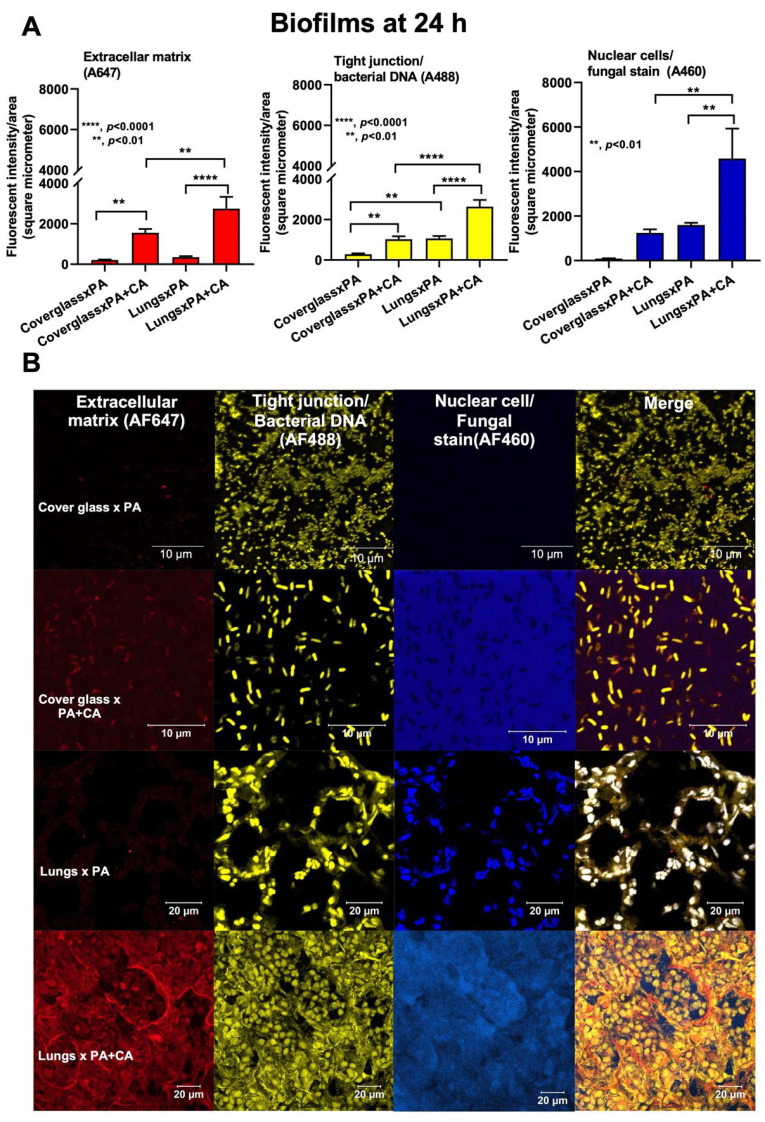
The characteristics of biofilms on abiotic glass slides and on biotic mouse lung tissue from Pseudomonas alone (PA), Pseudomonas with Candida (PA+CA), and control after staining for extracellular matrix (AF467, red), cell tight junction, bacterial DNA (AF488, yellow), and nuclei of host cells and fungi (AF460, blue), as indicated by fluorescent intensities (**A**), and the representative pictures (**B**) (independent experiments were performed in triplicate for biofilms on glass slides, and data of the in vivo biofilms were derived from *n* = 6–7 per group). Despite being the same color, the linear feature of the tight junction and the smaller size of fungal nuclei compared to the host are used for differentiation (see text).

**Table 1 ijms-23-09202-t001:** Quantitative real-time PCR primers.

Target Gene	Oligonucleotide Sequence (5-3)
Anti-sigma factor MucA (*mucA*)	F-GCGGATGAACTCGAGTTG R-CACTGACGGCGGATTGTT
Negative regulator for alginate biosynthesis MucB (*mucB*)	F-CTCCCTGTTGCTTTTGCTTG R-GATCTCATGGGTGGAGAAGC
GDP-mannose-6-dehydrogenase AlgD (*algD*)	F-AGGGCAACTGGACGGCTATC R-TGTGGTCGGCAATGAAGAAGA
Exopolysaccharide Psl (*pslA*)	F-ATGAACGCTCTGTTCGATTGTCCAC R-TCAAGCACTTGCACAGCAGACCT
*16S rRNA*	F-ACGCAACTGACGAGTGTGAC R-GATCGCGACACCGAACTAAT
Toll-like receptor 4 (*TLR4*)	F-CACAGACTTGCGGGTTCTAC R-AGGACCGACACACCAATGATG
Toll-like receptor 5 (*TLR5*)	F-ATTGCGTGTACCCTGACTCG R-TTGAACACCAGTCTCTGGGC
Toll-like receptor 9 (*TLR9*)	F-GTGACAGATCCAAGGTGAAGT R-CTTCCTCTACAAATGCATCACT
Interleukin-8 (*IL-8*)	F-TAGCAAAATTGAGGCCAAGG R-GGACTTGTGGATCCTGGCTA
Interleukin-13 (*IL-13*)	F-GCAATGGCAGCATGGTATGG R-AAGGAATTTTACCCCTCCCTAACC
Glyceroldehyde-3-phosphate dehydrogenase (*GADPH*)	F-GTGAAGGTCGGTGTCAACGGATTT R-CACAGTCTTCTGAGTGGCAGTGAT

## Data Availability

The data presented in this study are available on request from the corresponding author.

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
