# Peer review of "Rapid Synergistic Biofilm Production of Pseudomonas and Candida on the Pulmonary Cell Surface and in Mice, a Possible Cause of Chronic Mixed Organismal Lung Lesions"

_ijms, 2022, doi:10.3390/ijms23169202_

Round 1
Reviewer 1 Report
This work is an interesting source of knowledge, posing new questions for further research. the abstract and introduction are presented correctly. the results are well described, but the figures (despite the very good presentation of the results, some parts of the section could be equalized, the results belong to the discussion like: "Because i) the prominent biofilm production of PA+CA over PA alone might be due 99 to the different biofilm properties, and ii) the importance of several genes for Pseudomonas 100 biofilm production, especially the biofilms from polysaccharide synthesis locus (Psl) ver-101 sus alginate-mediated pathway [22,25], " the other way around, some parts of the disc are duplicate results - this should be avoided.
First of all, the data description in the way: "Notably, the incubation by Candida alone did not induce biofilms on the pulmonary cells (data not shown)."Additionally, both PA+CA and PA alone did not alter the number of pul-83 monary cells (data not shown) despite the Pseudomonas cytotoxicity [6]" , should be forbidden. please show you data in supplementary materials - this could be important to document your research properly.
Author Response
Reviewer 1
This work is an interesting source of knowledge, posing new questions for further research. The abstract and introduction are presented correctly. The results are well described, but the figures (despite the very good presentation of the results, some parts of the section could be equalized, the results belong to the discussion like: "Because i) the prominent biofilm production of PA+CA over PA alone might be due to the different biofilm properties, and ii) the importance of several genes for Pseudomonas biofilm production, especially the biofilms from polysaccharide synthesis locus (Psl) versus alginate-mediated pathway [22,25], " the other way around, some parts of the disc are duplicate results - this should be avoided.
ANS: We thank the reviewer for the comment and remove discussion from the result part and cut result from the discussion.
First of all, the data description in the way: "Notably, the incubation by Candida alone did not induce biofilms on the pulmonary cells (data not shown)."Additionally, both PA+CA and PA alone did not alter the number of pulmonary cells (data not shown) despite the Pseudomonas cytotoxicity [6]" , should be forbidden. please show you data in supplementary materials - this could be important to document your research properly.
ANS: We thank the reviewer for the comment and add the data (the incubation by Candida alone did not induce biofilms on the pulmonary cells (data not shown) as supplement figure 2. For “Additionally, both PA+CA and PA alone did not alter the number of pulmonary cells (data not shown) despite the Pseudomonas cytotoxicity [6]", we correct this statement as following with a supplement figures using MTT assay “Additionally, PA+CA and PA alone reduced the number of pulmonary cells as indicated by the MTT assay (more prominent in PA+CA than PA alone) (Supplement figure 1A) due to the Pseudomonas cytotoxicity”.

Reviewer 2 Report
The paper by Phuengmaung et al descibes the synergistic biofilm production by Pseudomonas and Candida in vitro and in vivo. Taking into account wide spread mixed fungal-bacterial biofilm-associated infections, the development of relevant models is an important challenge over the world. Pseudomonas-Candida biofilms are poorly described, that makes the research quiete inetersting. Overall the experimental design is relevant and adeqatelly described, and idea is publication-worth.
The reviewer thinks that the paper is suitable for consideration for acceptance with minor revisions.
Lines 44-54 Along with Pseudomonas aeruginosa and Candida albicans mixed infections, S.aureus - P.aeruginosa and S.aureus- C.albicans communities can be formed. The reviewer thinks that infarmation about these consortia also should be mentioned.
Line 83 Additionally, both PA+CA and PA alone did not alter the number of pulmonary cells. The microscopy on Fig 1 B shows that in presence of PA or PA+CA much less cells are present per field. Please correct.
Author Response
Reviewer 2
The paper by Phuengmaung et al describes the synergistic biofilm production by Pseudomonas and Candida in vitro and in vivo. Taking into account wide spread mixed fungal-bacterial biofilm-associated infections, the development of relevant models is an important challenge over the world. Pseudomonas-Candida biofilms are poorly described, that makes the research quite interesting. Overall the experimental design is relevant and adequately described, and idea is publication-worth.
The reviewer thinks that the paper is suitable for consideration for acceptance with minor revisions.
Lines 44-54 Along with Pseudomonas aeruginosa and Candida albicans mixed infections, S. aureus - P. aeruginosa and S. aureus - C. albicans communities can be formed. The reviewer thinks that information about these consortia also should be mentioned.
ANS: We thank the reviewer for the comment and add the information.
Line 83 Additionally, both PA+CA and PA alone did not alter the number of pulmonary cells. The microscopy on Fig 1 B shows that in presence of PA or PA+CA much less cells are present per field. Please correct.
ANS: We apologized for our mistake and changed it into “Additionally, PA+CA and PA alone reduced the number of pulmonary cells as indicated by the MTT assay (more prominent in PA+CA than PA alone) (Supplement figure 1A) due to the Pseudomonas cytotoxicity”.

Round 2
Reviewer 1 Report
The work has been properly improved. In my opinion, the discussion section lacks a description of therapeutic methods for bi-species biofilms. Interesting works on the topic:
https://www.frontiersin.org/articles/10.3389/fcimb.2020.550505/full
https://journals.asm.org/doi/10.1128/iai.00626-21
https://www.liebertpub.com/doi/full/10.1089/mdr.2021.0324
Author Response
Reviewer 1
The work has been properly improved. In my opinion, the discussion section lacks a description of therapeutic methods for bi-species biofilms. Interesting works on the topic:
https://www.frontiersin.org/articles/10.3389/fcimb.2020.550505/full
https://journals.asm.org/doi/10.1128/iai.00626-21
https://www.liebertpub.com/doi/full/10.1089/mdr.2021.0324
ANS: We thank the reviewer for the comment and added a paragraph on this topic with citation of these works as following “Interestingly, treatment of bacterial-Candida interspecies biofilms is challenging as the effect of antibacterial alone seems to be decreased in the dual-species biofilms com-pared with the biofilms from bacteria alone (Roszak M, 2022, 613). Meanwhile, in Pseu-domonas-Candida biofilms, a single treatment by an anti-fungal drug (fluconazole) alone synergistically with Pseudomonas on the biofilms in more effectively reduces Candida burdens partly through iron sequestration by P. aeruginosa(Hattab S,. Infect Immun. 2022 Apr 21;90(4):e0062621). These data indicate that anti-fungal might be beneficial in the treatment of bacterial-Candida interspecies biofilms. Moreover, the use of quorum quenching (quorum sensing blockage) or anti-virulence compounds (inhibitors against the virulent factors that are developed from the dual biofilms) are the interesting new strategic treatments of inter-species biofilms (Grainha T,. Front Cell Infect Microbiol. 2020 Nov 11;10:550505.). More studies on this topic are interesting.”